# Assessment of the Accuracy, Usability and Acceptability of a Rapid Test for the Simultaneous Diagnosis of Syphilis and HIV Infection in a Real-Life Scenario in the Amazon Region, Brazil

**DOI:** 10.3390/diagnostics13040810

**Published:** 2023-02-20

**Authors:** Daniela Cristina Soares, Luciano Chaves Franco Filho, Herald Souza dos Reis, Yan Corrêa Rodrigues, Felipe Bonfim Freitas, Cintya de Oliveira Souza, Giseli Nogueira Damacena, Nazle Mendonça Collaço Véras, Pamela Cristina Gaspar, Adele Schwartz Benzaken, Joana da Felicidade Ribeiro Favacho, Olinda Macedo, Maria Luiza Bazzo

**Affiliations:** 1Laboratory of Sexually Transmitted Infections, Bacteriology and Mycology Section, Evandro Chagas Institute (IEC), Ananindeua 67030-000, Brazil; 2Retrovirus Laboratory, Virology Section, Evandro Chagas Institute (IEC), Ananindeua 67030-000, Brazil; 3Institute of Scientific and Technological Communication and Information in Health, Oswaldo Cruz Foundation, Rio de Janeiro 21045-360, Brazil; 4Department of Diseases of Chronic Condition and Sexually Transmitted Infections, Ministry of Health, Brasilia 70723-040, Brazil; 5Fiocruz Amazônia, Manaus 69057-070, Brazil; 6AIDS Healthcare Foudation (AHF), Los Angeles, CA 90028, USA; 7Graduate Program in Pharmacy, Center for Health Sciences, Federal University of Santa Catarina, Florianópolis 88040-900, Brazil

**Keywords:** HIV, Syphilis, screening, diagnosis, duo test, SD bioline duo assay, sensitivity, specificity, kappa coefficient, Amazon

## Abstract

We field-assessed the accuracy, acceptability, and feasibility of the SD BIOLINE HIV/Syphilis Duo rapid diagnostic test in three groups: pregnant women, female sex workers (FSW), and men who have sex with men (MSM). Venous blood samples collected in the field were compared with the respective gold standard methods: SD BIOLINE HIV/Syphilis Duo Treponemal Test versus FTA-abs (Wama brand) treponemal laboratory test for syphilis, and SD BIOLINE HIV/Syphilis Duo Test versus the fourth generation Genscreen Ultra HIV Ag-Ag (Bio-Rad brand) laboratory test for HIV. From a total of 529 participants, 397 (75.1%) were pregnant women, 76 (14.3%) FSW and 56 (10.6%) MSM. Sensitivity and specificity parameters of HIV were 100.0% (95% CI: 82.35–100.0%) and 100.0% (95% CI: 99.28–100.0%), respectively. Sensitivity and specificity parameters found for TP antibody detection were 95.00% (95% CI: 87.69–98.62%) and 100.0% (95% CI: 98.18–100.0%), respectively. The SD BIOLINE HIV/Syphilis Duo Test showed high acceptability among participants (85.87%) and health professionals (85.51%), as well as easy usability by professionals (91.06%). The usability of the SD BIOLINE HIV/Syphilis Duo Test kit would not be a barrier to accessing rapid testing, if the product were incorporated into the list of health service supplies.

## 1. Introduction

The global pharmaceutical industry has produced several point-of-care health technologies aimed to improve communicable disease diagnosis, with emphasis on rapid solid-phase immunochromatographic assays for qualitative detection of antibodies and antigens. Since 2012, the Brazilian Ministry of Health has invested in rapid testing strategies, offering individual rapid tests for HIV and syphilis, combined with a structured process of acquisition, distribution, online technical training, and quality assessment by referral laboratories before implementation and availability in the healthcare system [1,2].

From the beginning of the epidemic in 1980s to June 2021, a total of 1,045,355 AIDS cases were reported in Brazil, including 381,793 new cases of HIV registered in 2021, with a detection rate of 14.1/100,000 inhabitants [3]. In 2020, 115,371 cases of acquired syphilis (detection rate of 54.5 cases/100,000 inhabitants), 61,441 cases of syphilis in pregnant women (detection rate of 21.6/1000 live births), 22,065 cases of congenital syphilis (with an incidence rate of 7.7/1000 live births), and 186 deaths from congenital syphilis (mortality rate of 6.5/1000 live births), were reported [4]. Aiming to enhance diagnosis and detection rates, the World Health Organization (WHO) prequalified the SD Bioline HIV/Syphilis Duo Test, making it the first simultaneous diagnostic point of care available for purchase by the public and private sectors, and recommended the application of the Duo rapid test in antenatal services and other testing sites, citing the simplification of acquisition process and advantages such as: reduction of storage space; simplification of training for healthcare professionals; use of a single finger prick; receiving test and treatment results in a shorter period; and trying to reduced unit cost for reagents compared to two single rapid tests for HIV and syphilis [5,6].

For HIV and syphilis, only rapid tests demonstrating sensitivity performance of ≥99.5% and ≥94.5%, and specificity performance of ≥99.0% and ≥93.0%, respectively, are acquired [3,4]. Further, operational performance criteria are also considered, which require the use of rapid tests presenting only a single reagent, storage at room temperature, execution in a maximum of four steps, reading in up to 30 min, and no laboratory experience requirement to perform the rapid test [7]. The SD Bioline HIV/Syphilis Duo Test results from laboratory prequalification revealed performance parameters of 100% (95%CI 98.2–100%) and 99.5% (95%CI 97.2–100%) on sensitivity and specificity for HIV, respectively, while for *Treponema pallidum* antibodies detection, the final sensitivity was 87% (95%CI 81.5–91.3%), with specificity of 99.5% (95%CI 97.2–100%) compared to reference (gold standard) essays [5].

Despite the aforementioned advantages, there is a lack of data on the duo test acceptance by service users in Brazil, especially in specific population segments where cases of HIV infection and syphilis has high incidence, such as men who have sex with men (MSM), female sex workers (FSW), and pregnant women in imminent risk for vertical transmission. Another relevant aspect, but less investigated, is the usability perception by healthcare professionals. The present work report results on the first survey on accuracy, acceptability and usability of the SD BIOLINE HIV/Syphilis Duo Test conducted in a real-life scenario in Brazil, allowing understanding the advantages and disadvantages of incorporating this health technology in the healthcare service and guarantee access to safe and quality diagnosis.

## 2. Materials and Methods

### 2.1. Study Design and Settings

This an epidemiological field-based, cross-sectional study, one which evaluated the accuracy, acceptability and usability performance of the SD BIOLINE HIV/Syphilis Duo Test for HIV and syphilis screening in the three different key populations—pregnant woman, MSM and FSW—in the city of Belém, capital of Pará State, Northern Brazil, from April to July 2021. The study design was based on a previously described protocol by the ProSPeRo Network [8].

Convenience sampling was employed considering the availability of individuals willing to participate in the study, supplies, and personal staff. Eligible participants were recruited at distinct sites: pregnant women attending to antenatal care routine services were recruited at three clinical sites, including Icoaraci, Bengui II and Paraiso dos Pássaros healthcare centers; MSM were recruited at Belém’s Testing and Counseling Center for STI’s, and female sex workers were recruited during a targeted action for testing and counseling at their working places (three nightclubs).

### 2.2. Participant Enrollment and Testing Procedures

Participants were invited and enrolled by a trained field researcher at each recruitment site, who explained the study purpose, testing procedures and applied an informed consent form (ICF) (Figure 1). General inclusion criteria consisted of participants 18 years and older, who provided informed written consent and were willing to be interviewed and tested. In particular for MSM, included individuals had to report engaging in sexual activity with another male in the last 12 months, and for FSW, engaging in sexual activity in exchange of payment or some other benefit in the last 12 months. Restriction regarding period of pregnancy or prenatal care were not applied for inclusion of pregnant women. Participants who refused to provide consent, who were not willing to be interviewed and tested, or who were under the influence of drugs or alcohol at the time of their participation in the study were excluded. The use of antimicrobials by participants who received prescribed treatment for syphilis or other infections in any period prior to study enrollment was recorded, but not used as exclusion criteria.

The evaluated kit consisted of the SD BIOLINE HIV/Syphilis Duo Test for HIV (Abbot, South Korea) is a solid phase immunochromatographic assay for the qualitative, simultaneous detection of antibodies to all isotypes (IgG, IgM, IgA) specific to HIV-1/2 and *Treponema pallidum* in human serum, plasma or whole blood. The manufacturer reports sensitivity and specificity values of 99.91% and 99.67% for HIV, and 99.67% and 99.72% for syphilis, respectively [5]. Each kit includes 25 essays, containing a vial of diluent solution, 25 pipettes, 70% alcohol wipes and an instruction form. The evaluated lancet was a 23 g safety lancet type, a different model from the one provided in the kit, because the only lancet accepted for use by the Unified Health System of Brazil (SUS) is of this type.

A fingerpick/whole blood sample was used to perform the SD BIOLINE HIV/Syphilis Duo Test according to the manufacturer’s instructions. Testing results were recorded in worksheets, counter-checked by a technician and a trained field researcher, reviewed online after photographic registration and sending via message app, and finally sealed in envelopes. In case of an invalid result, another test was performed. The test was reported as “Invalid” if the result of the repeated test was still invalid. In a case of discordant reading between the two independent evaluators, a third reader performed a final reading.

Results of this duo rapid antibody test under evaluation were not reported to participants. In addition, 5 mL of venipuncture whole blood was collected into an EDTA collection tube which was stored and then transported to the reference laboratory, in a temperature-controlled container within six hours of specimen collection.

Reference testing considered the national HIV and syphilis testing algorithm, which was performed at reference laboratories (STI’s laboratory, Bacteriology and Mycology section; and Retrovirus laboratory, Virology Section—Evandro Chagas Institute, Ministry of Health) by a laboratory staff, who were blinded to prior test results and clinic information of the participants. Standard reference test results were reported to participants, who received an issued report with HIV and syphilis tests results. All the documents were filed and delivered sealed in the health units where the participants were recruited, except for the FSW. For FSW, the tests results were filed at the fixed healthcare unit belonging to the “Consultório na Rua/Street Clinic strategy” and at the nightclubs in which were recruited. Pregnant women and MSM participants received a text message via messaging application and/or an email with the conclusion and research achievements, except for the FSW, as they did not provide any phone number or e-mail address. Finally, a seminar aiming to present the project results was held on 17 September 2021 at the Evandro Chaga Institute (IEC/PA). The event was directed to SESMA (Municipal Secretary of Health), basic healthcare units and CTA managers and staff, as well as to GEMPAC (Group of Female Prostitutes From the State of Pará) members, a non-governmental association representing the FSW.

### 2.3. Acceptability and Usability Survey

To determine the acceptability of the investigated SD BIOLINE HIV/Syphilis Duo Test, a trained research assistant conducted a one-on-one interview for completing a socio-demographic questionnaire with each participant at field study sites. Acceptability among participants was determined by evaluating preference for rapid POC testing or laboratory testing; preference for rapid duo test or single test for each disease; willingness to wait for results and how much time they would be willing to wait for these results, and finally which of the tests would stimulate testing for HIV/Syphilis by the participants.

The usability of the SD BIOLINE HIV/Syphilis Duo Test was evaluated by health professionals from nine different services linked to the Primary Health Care strategy in the city of Belém. Health professionals contacted in advance by research staff were enrolled after an explanation of the study’s purpose, and gave consent by signing the specific ICF. The research staff provided the kit and supplies for evaluation, together with a video with guidelines for test execution. Then, the healthcare professionals organized themselves into pairs, executed the test in each other, and finally, filled out the survey usability form.

### 2.4. Data Analysis

A data entry form was created (Google Form) for each key population investigated, where questionnaires data and test results (laboratory and rapid test) were uploaded. All collected data was counter-checked and submitted to quality control daily. From this database, proportions were calculated for categorical variables by using excel.

Contingency tables were constructed in order to calculate sensitivity, specificity, positive predictive value, negative predictive value and accuracy performance. The rapid test results were compared with the respective reference (gold standard) methods: SD BIOLINE HIV/Syphilis Duo Treponemal Test versus FTA-abs (Wama brand) treponemal laboratory test for syphilis, and SD BIOLINE HIV/Syphilis Duo Test versus the 4th generation Genscreen Ultra HIV Ag-Ag (Bio-Rad brand) laboratory test for HIV. Confidence intervals (CIs) of 95% were considered for each estimate. The calculations were performed using the software MedCalc (https://www.medcalc.org/, accessed on 19 July 2022). The kappa index was calculated, to compare concordance between the reference tests results and the SD BIOLINE HIV/Syphilis Duo Test, using the software VassarStats (http://vassarstats.net/, accessed on 20 July 2022). The results were compared to performance values described by the manufacturer and prequalification values by the World Health Organization and the Brazilian Ministry of Health manuals for diagnosis.

### 2.5. Ethical Considerations

The present study had approval granted by the Research and Ethics Committee of the Evandro Chagas Institute (No. 19146919.3.0000.0019, approval date: 27 August 2019) and the Pan American Health Organization (No. PAHOERC-2019-08-0059, approval date: 27 August 2019). Consent was also provided by the Permanent Education Center of the Health Department of Belém (Date: 25 July 2019). Each participant was invited and voluntarily enrolled by providing informed consent. Study data and blood samples were identified by unique identification codes, aiming to protect data and maintain confidentiality.

## 3. Results

### 3.1. SD Bioline HIV/Syphilis Duo Test Kit Performance

From a total of 529 participants, 397 (75.1%) were pregnant women, 76 (14.3%) FSW and 56 (10.6%) MSM. For the syphilis, rapid Duo testing revealed a total of 76 (14.3%) reactive participants, including 29 (13.7%) pregnant women, 18 (32.1%) MSM and 29 (38.1%) FSW; while on the reference testing by FTA-Abs, 80 (15.1%) of participants were reactive, including 31 (7.8%) of pregnant women, 18 (32.1%) MSM and 31 (40.7%) FSW. For HIV, the rapid Duo testing and reference standard essay presented concordant results with 19 (3.6%) reactive participants, including four (1.0%) of pregnant women, 11 (19.6%) MSM and four (5.2%) FSW.

The concordance performance of the rapid Duo test with the reference test demonstrated sensitivity and specificity parameters of 100.0% (95% CI: 82.35–100.0%) and 100.0% (95% CI: 99.28–100.0%), respectively, for HIV (kappa coefficient: 1.00 [95% CI, 1.00–1.00]). As for the differences, sensitivity and specificity parameters found for TP antibody detection were of 95.00% (95% CI: 87.69–98.62%) and 100.0% (95% CI: 98.18–100.0%), respectively (kappa coefficient: 0.9705 [95% CI: 0.9417–0.9993]) (Table 1 and Table 2).

### 3.2. Pregnant Women, MSM and FSW Acceptability

Among the studied groups, over 50% of both pregnant women and MSM would choose the rapid test over the laboratory test. Regarding the use of the rapid test to detect one or two simultaneous infections (duo), most participants would prefer the duo test. As for the waiting time, despite not exceeding 50% of the audience in each group, most participants would like to receive the test results swiftly (30 min to 1 h) (Table 3).

### 3.3. Healthcare Professional’s Acceptability

Among the interviewed healthcare professionals, the rapid Duo test was accepted by 85.51%, with 5.80% responding not preferring the simple or Duo test, as they have the ability to handle both tests. In relation to healthcare professionals’ opinion on the test acceptance by the target public, the majority responded that all groups would agree to be tested using the rapid Duo test (89.86%). According to the professionals, the group with the greatest refusal could be the FSW, with a percentage of 4.35% (Table 4).

### 3.4. Feasibility of the Duo Rapid Test among Healthcare Professionals

A total of 138 healthcare professionals answered the questionnaire regarding professional training, operational characteristics of the Duo test kit, and usability suggestion by the target public. Among the healthcare professionals, the majority have graduated from high school (57.24%) and were professionally trained as a nursing technician (51.44%). Also, most of the professionals classified the test as easy (91.06%) to operate and apply, and specifically reported difficult in the use of the provided pipette for blood collection; “interpretation of results” was the characteristic with the best acceptability among healthcare professionals (95.65%) (Table 5).

## 4. Discussion

The present study reports one of the first investigations conducted in Brazil on the accuracy, acceptability and usability of a rapid test for simultaneous diagnosis of HIV and syphilis. The SD BIOLINE kit, the only kit available in the country, during the pandemic period, was evaluated, given the limitations imposed by the shortages during the pandemic. However, there are products with the same purpose registered in Brazil whose manufacturers are MedLevensohn^®^ (Rio de Janeiro, Brazil), Eco Diagnóstica (Minas Gerais, Brazil), Lumiradx (São Paulo, Brazil) and Bio Manguinhos (Rio de Janeiro, Brazil).

Understanding the epidemiological scenario is highly important for guiding the prioritization of prevention measures, such as rapid testing. It is also known that, despite the availability of diagnostic tools for HIV and syphilis in the market and in the healthcare system, other factors are pointed out as barriers to access to testing. Regarding STIs, and from the perspective of users, especially key populations such as MSM and FSW, prejudice and stigma, acceptance of sexuality, fear of the result, and lack of information are highlighted as the main barriers to accessing testing [9]. The present study revealed that offering a simultaneous rapid test for HIV and syphilis diagnosis would not drive users away from health services given that most prefer rapid testing over laboratory testing (Table 3). Among the pregnant women, MSM, and FSW groups, wide acceptance of the rapid Duo test was observed (Table 3). In the USA, 28% of patients screened for HIV in an emergency medical service believed that rapid testing was less or much less accurate than conventional testing, reinforcing the need for acceptability studies prior to the decision to implement new diagnostic devices in healthcare services [10]. A study conducted in Peru concluded high acceptability of rapid testing among transgender women (98.8%), however, in Argentina, only 60.7% of this public preferred simultaneous HIV and syphilis diagnostic testing [11]. Regarding the waiting time, we observed that most of participants prefer to receive the results as quickly as possible and in less than an hour. Flores [12], demonstrated that the implementation of simultaneous screening for HIV/syphilis testing in a health service in Peru resulted in 52% completely satisfied patients and 48% satisfied with the improvements in care processes.

The acceptability of rapid tests was not a consensus among healthcare professionals, as in the past there has been enormous resistance during their application in healthcare services in Brazil. Many professionals did not trust the results expressed on the test strip, and preferred to wait for results issued by the reference laboratories, and by means of referenced methods, for the purposes of clinical management. However, scientific evidence generated on the accuracy of rapid tests in comparison to gold standard tests, and the adoption of minimum parameters of sensitivity and specificity, mitigated this resistance. Such scenario was also observed in the present study (Table 4).

In addition, practitioners in service demonstrated a preference to use the duo test over the single test, mainly due to its ease of application and reduction supplies (data not shown). The 138 volunteer healthcare professionals enrolled in this study reported that the SD BIOLINE was easy to perform (Table 5). The use of the collection pipette was the most difficult feature described, but classified as such only by a small proportion of participants (Table 5). Although most healthcare professionals are qualified to apply the rapid test, some reported not using it frequently or had started working with this function only recently, which may explain their lower skill with the collection pipette (data not shown). Other participants also stated that users of health services would accept the simultaneous rapid test for diagnosis of HIV and syphilis, which was ratified by the users’ responses (Table 3).

Despite specific strategies for healthcare professionals’ qualification on rapid tests’ application in the context of STIs, Brazil adopts such technologies for several other infections. At the time of this study, many professionals had also been trained to perform rapid tests for COVID-19 detection, which may contribute to the high rate of ease performing the SD BIOLINE test. Even though, the reading of results being one of the features rated as easiest by the professionals, such finding is limited as the ProSPeRo Network protocol for usability investigations recommend that each professional perform at least 50 rapid tests before answering the questionnaire [8]. During the field application of the duo test, the project staff reported that variable color intensity in the syphilis band resulted, a fact also observed by Olugbenga [13], Bristow [14,15] and Heuvel [16]. This shows the need for training of positive test readings by professionals despite the easy classification indicated by them in this study, which may aid to avoid interpretation of erroneous results.

The observation of rapid test performance in a real-life scenario is another extremely relevant criteria for its incorporation in healthcare systems. The rapid tests for syphilis presented a sensitivity of 95%, a specificity of 100% and an accuracy of 99.25% (Table 1). For HIV, all measures were 100% in all groups (Table 2), which are above the sensitivity and specificity criteria recommended by the Brazilian Ministry of Health and WHO pre-qualifications [17]. Although the kappa value indicated almost-perfect agreement with the reference test, the results of the group of pregnant women for syphilis showed a sensitivity of 93.55%, as well for the FSW group, which is slightly below the established criteria of 94.5% [17]. These values are higher than showed by WHO in prequalification report for the case of antibodies to *Treponema pallidum*, with 87.0% (95% CI: 81.5–91.3%) for final sensitivity and the final specificity of 99.5% (95% CI: 97.2–100%) compared to the reference assays. Until now, the minimal sensitivity as specificity accepted by the Ministry of Health of Brazil was the same for single and for duo tests, and for all types of specimens (e.g., serum, plasma, or whole blood). It would be important to review these parameters considering the methodology differences, different specimens’ type and other aspects, such as the increase of access and acceptability.

Studies conducted by Black et al. [18], in South Africa, also showed low sensitivity of the SD BIOLINE for the diagnosis of syphilis in FSW from the finger stick whole blood sample. Studies with pregnant women in Zambia and Vietnam also revealed low sensitivity for syphilis diagnosis from SD BIOLINE finger stick whole blood sample [19,20]. Assays of the SD BIOLINE with serum and plasma samples also showed low sensitivity for syphilis diagnosis performed in pregnant women and MSM in the US, South Sudan and Zimbabwe [21,22,23]. SD BIOLINE was 100% sensitive from plasma of pregnant women in Uganda [24]. Despite the low sensitivity shown for syphilis diagnosis, such studies provided a satisfactory specificity. The specificity values for pregnant women and FSW groups are within the limit recommended by the Brazilian Ministry of Health (93% specificity). In areas of high prevalence rates of a disease, the higher the test specificity, the higher the positive predictive value are found, therefore, higher the probability of disease in cases of positive result. It is worth mentioning that some factors can interfere with the accuracy of a diagnostic test, such as execution failures, the use of the incorrect volume of buffer or sample, reading the test result at the incorrect time, incorrect interpretation of the result, difficulty in interpreting weakly reagent bands, etc.

In this study, the field phase included double-checking of rapid test results. The survey also asked the volunteers about some factors already reported as possible causes of false-reactive results in rapid tests, such as autoimmune diseases, liver disease, pregnancy, recent H1N1 vaccination, hemodialysis patients, and patients on interferon therapy, among others. However, none of these factors seems to be related to the four false-negative cases for syphilis (Table 1). These were serological scars. In relation to HIV, all indices of all specimens tested were 100% compared to the gold standard, meeting the criteria defined by the ministry and PAHO [17], which is 99.5% for sensitivity and 99.0% for specificity. In addition, the syphilis results for the MSM group showed all rates at 100%, meeting the recommended criteria [7].

## 5. Conclusions

This study concludes that the accuracy, acceptability and usability of the SD BIOLINE HIV/Syphilis Duo Test kit would not be barriers to access to rapid testing, if the product were incorporated into the list of inputs of the health services.

## Figures and Tables

**Figure 1 diagnostics-13-00810-f001:**
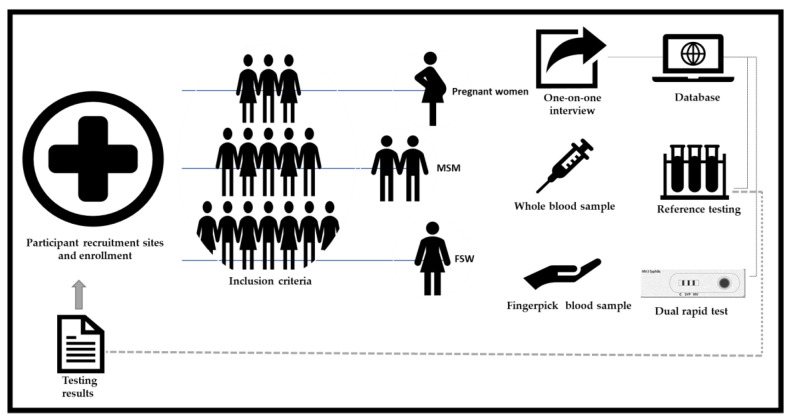
Key populations enrollment and testing procedures.

**Table 1 diagnostics-13-00810-t001:** Concordance performance of the rapid Duo test with the reference test for Syphilis.

*n* = 529	Treponemal Test X Rapid Test							
	Total		Pregnant Women		MSM		FSW	
True Positive	76		29		18		29	
True Negative	449		366		38		45	
False Positive	0		0		0		0	
False Negative	4		2		0		2	
	Value	95% CI	Value	95% CI	Value	95% CI	Value	95% CI
Sensitivity (%)	95	87.69–98.62	93.55	78.58–99.21	100	81.47–100	93.55	78.58–99.21
Specificity (%)	100	99.18–100	100	99–100	100	90.75–100	100	91.13–100
PLR	0.00	-	0.00	-	0.00	-	0.00	-
NLR	0.05	0.02–0.13	0.06	0.02–0.25	0.00	-	0.06	0.02–0.25
PPV (%)	100	-	100	-	100	-	100	-
NPV (%)	99.12	97.74–99.66	99.46	97.95–99.86	100	-	95.24	83.96–98.71
Accuracy (%)	99.25	98.08–99.79	99.50	98.19–99.94	100	93.62–100	97.18	90.19–99.66
	Total		Pregnant women		MSM		FSW	
Observed Kappa	0.9705		0.9652		1		0.9364	
	SE	95% CI	SE	95% CI	SE	95% CI	SE	95% CI
Method 1	0.0147	0.9417–0.9993	0.0246	0.917–1	0	1	0.0444	0.8494–1
Method 2	0.0147	0.9417–0.9993	0.0246	0.917–1	0	1	0.0444	0.8496–1

MSM: Men who have Sex with Men; FSW: Female sex workers; PLR: Positive Likelihood Ratio; NLR: Negative Likelihood Ratio; PPV: Positive Predictive Value; NPV: Negative Predictive Value; CI: Confidence Interval; SE: Standard Error.

**Table 2 diagnostics-13-00810-t002:** Concordance performance of the rapid Duo test with the reference test for HIV.

*n* = 529	Gold Standard X Rapid Test							
	Total		Pregnant Women		MSM		FSW	
True Positive	19		4		11		4	
True Negative	511		393		45		72	
False Positive	0		0		0		0	
False Negative	0		0		0		0	
	Value	95% CI	Value	95% CI	Value	95% CI	Value	95% CI
Sensitivity (%)	100	82.35–100	100	39.76–100	100	71.51–100	100	39.76–100
Specificity (%)	100	99.28–100	100	99.06–100	100	92.13–100	100	95.07–100
PLR	0.00	-	0.00	-	0.00	-	0.00	-
NLR	0.00	-	0.00	-	0.00	-	0.00	-
PPV (%)	100	-	100	-	100	-	100	-
NPV (%)	100	-	100	-	100	-	100	-
Accuracy (%)	100	99.31–100	100	99.07–100	100	93.62–100	100	95.32–100
	Total		Pregnant women		MSM		FSW	
Observed Kappa	1		1		1		1	
	SE	95% CI	SE	95% CI	SE	95% CI	SE	95% CI
Method 1	0	1	0	1	0	1	0	1
Method 2	0	1	0	1	0	1	0	1

MSM: Men who have Sex with Men; FSW: Female sex workers; PLR: Positive Likelihood Ratio; NLR: Negative Likelihood Ratio; PPV: Positive Predictive Value; NPV: Negative Predictive Value; CI: Confidence Interval; SE: Standard Error.

**Table 3 diagnostics-13-00810-t003:** Acceptability of the HIV-syphilis rapid diagnostic test among pregnant women, MSM and FSW.

		Rapid or Laboratory Test Preference			Duo or Single Test Preference			Time Willing to Wait for Results				
Pregnant Women		Rapid	Lab	Any Test	Duo	Single	Any Test	30 min	1 h	2 h	Wait any Time	Return Other Day
18–24	176	52.27	21.59	26.14	82.95	14.20	2.84	42.61	22.73	5.68	18.75	10.23
25–29	108	52.78	22.22	25	87.96	11.11	0.93	48.15	26.85	2.78	15.74	6.48
30–49	112	50.89	24.11	25	91.07	7.14	1.79	46.43	21.43	6.25	16.07	9.82
50+	1	100	0	0	100	0	0	0	0	0	100	0
Total	397	52.14	22.42	25.44	86.65	11.34	2.02	45.09	23.43	5.04	17.38	9.07
MSM												
18–24	18	44.44	50	5.56	100	0	0	50	16.67	5.56	27.78	0
25–29	15	46.67	53.33	0	93.33	6.67	0	53.33	26.67	20	0	0
30–49	22	77.27	18.18	4.55	90.91	9.09	0	63.64	4.55	4.55	27.27	0
50+	1	100	0	0	100	0	0	0	0	0	100	0
Total	56	58.93	37.50	3.57	94.64	5.36	0	55.36	14.29	8.93	21.43	0
FSW												
18–24	27	48.15	22.22	29.63	59.26	33.33	7.41	44.44	7.41	14.81	29.63	3.7
25–29	17	47.06	17.65	35.29	82.35	11.76	5.88	47.06	23.53	17.65	11.76	0
30–49	22	22.73	50	27.27	86.36	9.09	4.55	36.36	27.27	9.09	27.27	0
50+	10	30	40	30	90	10	0	30	10	10	50	0
Total	76	38.16	31.58	30.26	76.32	18.42	5.26	40.79	17.11	13.16	27.63	1.32

**Table 4 diagnostics-13-00810-t004:** Acceptability of the HIV-syphilis rapid diagnostic test among healthcare professionals.

Test Type Preference		
	Quantity (*n*)	Percentage (%)
Duo	118	85.51
Single	12	8.70
Both/Any of them	8	5.80
**Would any user stop testing if there was only the DUO TEST for the simultaneous diagnosis of Syphilis and HIV?**		
No, all groups would agree to be tested	124	89.86
Yes, FSW	6	4.35
Yes, heterosexuals	4	2.90
Yes, MSM	3	2.17
Yes, LGBTQI+	1	0.72
Total	138	100

**Table 5 diagnostics-13-00810-t005:** Operational feasibility of the HIV-Syphilis rapid diagnostic test among healthcare professionals.

Operational Features	Easy	Reasonably Difficult	Difficult
Kit Instruction	124 (89.85%)	12 (8.70%)	2 (1.45%)
Package opening	126 (91.30%)	10 (7.25%)	2 (1.45%)
Use of the collection pipette	110 (79.71%)	18 (13.04%)	10 (7.25%)
Use of the piercing lancet	130 (94.20%)	6 (4.35%)	2 (1.45%)
Use of buffer solution	132 (95.65%)	5 (3.62%)	1 (0.72%)
Interpretation of results	132 (95.65%)	6 (4.35%)	0
Average of percentages (Total = 138)	91.06%	6.88%	2.05%

## Data Availability

All relevant data is presented within the manuscript.

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
