# Peer review of "Assessment of the Accuracy, Usability and Acceptability of a Rapid Test for the Simultaneous Diagnosis of Syphilis and HIV Infection in a Real-Life Scenario in the Amazon Region, Brazil"

_diagnostics, 2023, doi:10.3390/diagnostics13040810_

Round 1

Reviewer 1 Report

This is an excellent paper, ready for publication. I think your methodology is well chosen and executed. The use of various acceptance surveys strengthens the importance of the study and gives it a context that makes it relevant to practicing clinicians. 

Two small points:

1. You mention in the discussion (lines 245 - 250) that the SD BIOLINE test has various competitors already available in Brazil. Acceptance of this test in relation to the others will have an economic component (price!) in this resource-limited setting. Especially as Brazil attempts to rebuild its health and especially its STI disease infrastructure, economics would be a major deciding factor in choosing instruments that equal technical qualifications. This is just a thought.

2. Your English is generally excellent. However, a close additional reading will pick up a number of grammatical and word meaning errors. E.g. two uses of "thoughts" instead of "though".  

Author Response

We appreciate your availability to improve this scientific publication, and follows addressed the comments:

Comments and Suggestions for Authors – 1

This is an excellent paper, ready for publication. I think your methodology is well chosen and executed. The use of various acceptance surveys strengthens the importance of the study and gives it a context that makes it relevant to practicing clinicians. 

 Two small points:

  1. You mention in the discussion (lines 245 - 250) that the SD BIOLINE test has various competitors already available in Brazil. Acceptance of this test in relation to the others will have an economic component (price!) in this resource-limited setting. Especially as Brazil attempts to rebuild its health and especially its STI disease infrastructure, economics would be a major deciding factor in choosing instruments that equal technical qualifications. This is just a thought.

Reply: The results presented in this paper contributed to the incorporation of rapid tests for the simultaneous diagnosis of syphilis and HIV by the Brazilian Ministry of Health in 2021. Certainly, the supply of products from different manufacturers for the simultaneous diagnosis of syphilis and HIV through rapid tests in Brazil will contribute to broad competition and reduced procurement costs. However, with the budget cuts imposed by the previous government for actions against STIs, the tests were not purchased in 2022. In 2023, under a new government, we are excited and believe in the resumption of public health actions in Brazil.

  1. Your English is generally excellent. However, a close additional reading will pick up a number of grammatical and word meaning errors. E.g. two uses of "thoughts" instead of "though".  

Reply: We appreciate the comments and corrections were verified as requested. In addition, MDPI staff will perform the final English proofreading.

Reviewer 2 Report

Whereas the test for syphilis showed good sensitivity and specificity, it failed to recognize that TP tests may stay positive long after treatment.  This needs to be followed with an RPR to document active v inactive disease

Author Response

Comments and Suggestions for Authors – 2

 Whereas the test for syphilis showed good sensitivity and specificity, it failed to recognize that TP tests may stay positive long after treatment.  This needs to be followed with an RPR to document active v inactive disease

Reply: On average, every two years, the Brazilian Ministry of Health updates its Clinical Protocol of Therapeutic Guidelines for STIs. In this document, there are guidelines for clinical management when faced with a reactive rapid syphilis test to distinguish active disease from inactive disease. The guidelines include epidemiological and clinical investigation of patients and their sexual partners, history of penicillin treatment, among others. However, the enormous challenge in keeping health professionals trained and updated on clinical management recommendations for syphilis is well known, as there is a high turnover of professionals in public health in Brazil. The Brazilian Ministry of Health offers distance learning courses on prevention, diagnosis, and treatment; promotes technical meetings, and articulates dissemination actions with the professional class councils of physicians, nurses, pharmacists, biomedical doctors, etc., to disseminate knowledge.